# Bonding of Steel Bars in Concrete with the Addition of Carbon Nanotubes: A Systematic Review of the Literature

**Elvys Dias Reis** *[ID], **Heron Freitas Resende**, **Péter Ludvig** [ID], **Rogério Cabral de Azevedo** [ID],
**Flávia Spitale Jacques Poggiali** [ID] and **Augusto Cesar da Silva Bezerra** [ID]

Federal Center for Technological Education of Minas Gerais, Belo Horizonte 30421-169, Minas Gerais, Brazil
* Correspondence: elvysreis@yahoo.com.br

**Abstract:** Advances and innovations in science and engineering have been increasingly supported by nanotechnology, and the modification of cementitious materials by nanoengineering is an expanding field. With this perspective, this paper aims to elucidate the behavior of steel bars in concrete with the addition of carbon nanotubes (CNTs) as a function of the characteristics of the cement-based material, the dispersion techniques and dosage of CNTs, the bond tests and specimen geometry, and the rebar characteristics. To reach this proposed goal, the ProKnow-C methodology was applied to select the most relevant publications from the last ten years, and then seven articles were fully analyzed. The results of the present systematic review of the literature revealed both consolidated knowledge and gaps to be filled in future research, as the need to study the chemical effect of adding these nanomaterials for improving steel–concrete adhesion, the bonding of thin bars in concrete, and the real influence of anchorage length on the steel–concrete bond, regardless of the use of CNTs, is vital.

**Keywords:** bond strength; reinforced concrete; cement-based materials; pull-out test

## 1. Introduction

Innovations in science and engineering are progressively developing on several fronts, among which those provided by nanotechnology stand out [1]. In this sense, one of the materials that is increasingly investigated because of its potential to improve the performance of Portland cement-based materials is the carbon nanotube (CNT) [2], understood as a set of carbon structures that, after being synthesized, obtain a cylinder shape on a nanometric scale, then classified as single-walled (SWCNT) or multi-walled (MWCNT) based on the number of cylinders formed [3]. The elastic modulus of this material can be about six times greater than that of steel and its tensile strength can reach about 150 GPa [4,5]. With an aspect ratio (length/diameter) greater than any other known material, CNTs have been considered ideal candidates for application in composites, being potential controllers of micro-crack propagation [6].

In the context of civil construction, much research has been developed in Brazil and worldwide since the discovery of the CNT, in 1991, to investigate the best methods and contents for its use in cementitious materials, as well as the benefits in the performance of mechanical properties and durability of mortars and concretes with the addition of this material. However, this area has gained more prominence with some works developed in the last decade [7–11]. Although CNTs themselves have excellent properties, when incorporated into cementitious composites they do not always guarantee good performance due to two main limitations: (i) their difficulty of dispersion, mainly due to their hydrophobic nature and strong van der Waals forces between them [12]; and (ii) a weak interfacial interaction between them and the matrix [13]. Thus, it is noted that the results concerning the mechanical properties and durability of cementitious composites with the incorporation of CNT are divergent, as reported by several researchers [14–16].

To obtain a better dispersion in cementitious matrices, Ladeira et al. [17] patented a process for the synthesis of CNT directly on a Portland cement clinker using the CVD (chemical vapor deposition) method, enabling the industrial-scale production of a nanostructured cement. The use of this material showed significant enhancement in the mechanical behavior of cement mortars [18]. In order to obtain similar results, Rocha and Ludvig [19] performed tests with cement pastes incorporating CNT at levels of 0.10% and 0.05% by weight of cement after a dispersion process of CNTs on the surface of cement grains, which improved some properties, such as compressive strength and flexural tensile strength, fracture energy, and fracture toughness, suggesting the role of CNT as crack propagation controller. In addition, the authors concluded that the presence of CNT contributes to the reduction of pore volume, increases the density of cement pastes, and acts as nucleation sites for cement hydration products. Other authors have subsequently studied the microstructure of CNT concrete [20] and its mechanical properties and failure mechanisms when subjected to high temperatures [21].

The works mentioned above demonstrated that CNTs can be used to obtain more resistant and durable cementitious materials. However, because of their high innovation potential, there are still gaps regarding their use in Portland cement composites that need to be filled, such as the study of bonding between steel and CNT concrete, an essential mechanism for the use of reinforced concrete structures [22]. Since the bonding characteristics of reinforcement bars and concrete are influenced—besides many other characteristics—by concrete strength, CNT addition has an inherent effect on the bonding behavior due to the enhancement of mechanical properties. The study of bond behavior between steel bars and concrete can be performed through different tests, among which the pull-out test, standardized by RILEM-CEB RC6 [23], the beam flexural test, standardized by RILEM-CEB RC5 [24], and the confined bars test, standardized by the Brazilian standard NBR 7477 [25], deserve special attention. Considering these three tests, the pull-out is the most widely used and the least laborious [26].

Arel et al. [27] experimentally investigated the effect of the cover thickness and curing time on the steel–concrete bond strength, using the pull-out test [23], in concretes with eight levels of compressive strength. The authors noted the improvement in bond strength with increasing compressive strength, cover thickness, and curing time. In the following year, Pop et al. [28] compared the adherence to self-compacting concrete (SCC) with concrete compacted with a vibrator. Through the pull-out test, they observed higher bond strength and less slippage at the same load level in SCC specimens, and found, considering the influence of the bar diameter, similar behavior in both concretes. Some years later, Garcia-Taengua et al. [29] also performed pull-out tests to study the bond behavior of steel fiber-reinforced concrete. Normal strength concrete (between 20 and 50 MPa), steel bars with diameters between 8 and 20 mm, cover between 30 mm and 5 times the diameter of the rebar, fiber content up to 70 kg/m$^3$, and fibers with different lengths were used by the authors. The results revealed a very limited effect of fiber content on adherence, and that the larger the bar diameter and the compressive strength of concrete, the higher the bonding, which is consistent with the literature.

The bond behavior of rebars in concrete is governed by three different types of interaction: mechanical interlock, frictional resistance, and chemical bond. The addition of CNTs to concrete may interfere predominantly with the last two phenomena. It was already shown that this addition enhances significantly the mechanical characteristics of the cement-based matrix. According to the previously presented influence of concrete strength on the bond between concrete and reinforcement bars, a positive effect on the bond strength with the addition of CNT is also expected.

In this context, to the best of the authors' knowledge, there are few papers in the literature that address the influence of CNTs' addition on the adherence of steel bars with cementitious materials. Therefore, this manuscript aims to elucidate the state-of-the-art in this subject, considering the cement-based material, the characteristics, dispersion techniques and dosage of CNTs, the bond tests, specimen geometry, and the rebar char-

acteristics, to answer the following questions: (i) Is the bond behavior of steel bars well known in all types of concrete? (ii) Which type of CNT is more feasible to incorporate into concrete, MWCNT or SWCNT? (iii) Are powdered CNTs employed more often than CNTs in aqueous suspension? (iv) What is the ideal CNT dispersion technique? (v) What is the ideal dosage? (vi) What is the most appropriate test to study the steel–concrete bond? (vii) What is the most appropriate test to study the standard? (viii) How often are ribbed and smooth steel bars used in the study of steel–concrete adhesion? (ix) What is the range of steel bar diameters that should be further investigated? (x) What is the range of anchorage length that should be further investigated. For this, a systematic review of the literature was performed, to discuss already consolidated results and gaps in the science to be filled in future research.

## 2. Materials and Methods

A systematic review of the literature was performed through the ProKnow-C (knowledge development process—constructivist) methodology [30], which guarantees a better organization of the information, identifying consolidated aspects, knowledge gaps, and the main authors concerning the theme in question [31]. Figure 1 summarizes the main stages of this methodology.

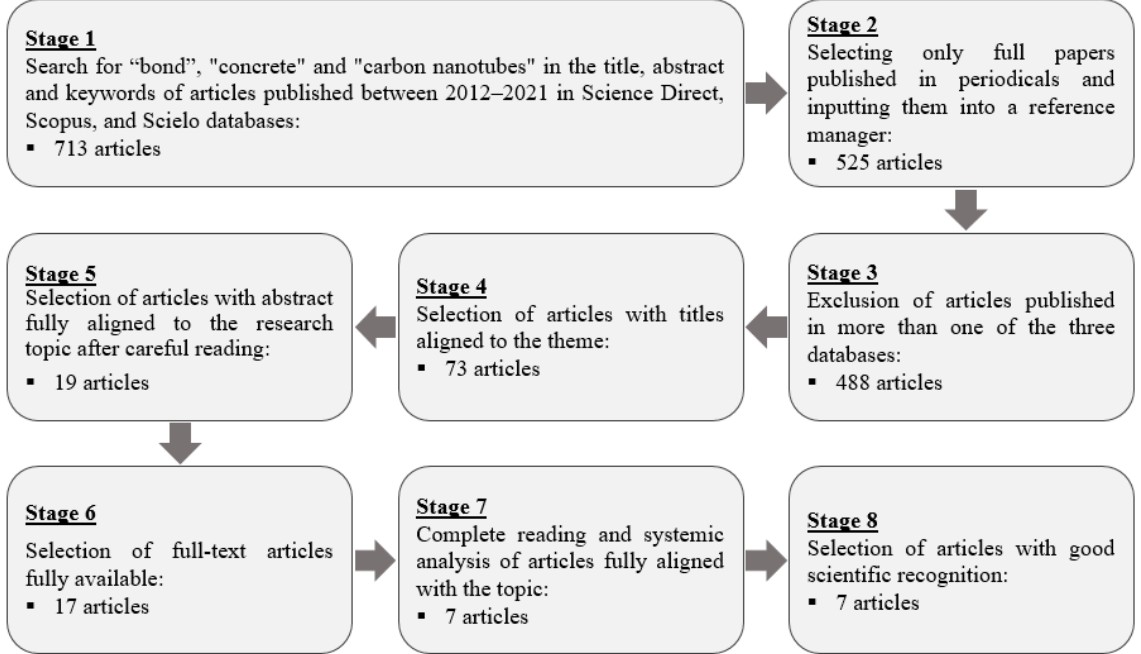

**Figure 1.** Stages of a systematic review of the literature using ProKnow-C.

The definition of the time frame and the databases indicated in the first step are justified, respectively, by the need to include the most recent research on the subject and by the availability of full texts with good scientific recognition. From the second to the sixth stages, after checking the established criteria, 17 articles were selected for full reading. Considering these articles, it was found that, although ten of them addressed the research topic, they did not deeply study the adherence of steel bars in CNT concrete, focusing on topics such as mechanical and durability properties, among others. Thus, only seven were fully aligned with the theme and were used in the systematic analysis in the next stage. Finally, it was verified that these seven articles ensured excellent scientific recognition in the portfolio [32].

After the complete reading of the articles, a systematic analysis was carried out to identify gaps and highlights in the knowledge around the researched theme, which was done using an approach called lenses [30]. In this work, the research lenses were

established considering the most relevant aspects for surveying the state-of-the-art of steel bar adherence in CNT concrete, as follows: concrete type, dispersion technique, CNT content, test and standard used, bar type, bar diameter, and bar anchorage length.

## 3. Results

### 3.1. Selected Bibliography

Table 1 lists the articles that make up the bibliographic portfolio selected through the ProKnow-C methodology and their main information: authors, journal, year of publication, and number of citations [33]. This is a very recent subject, with the first manuscript being published just over three years ago and the most recent one less than a year ago. It is also noteworthy that, considering the publication year, the articles present a good number of citations.

**Table 1.** Details of the articles selected applying the ProKnow-C methodology.

| Reference | Authorship | Title | Journal | Year | Citations |
|---|---|---|---|---|---|
| [34] | Hawreen, A.; Bogas, J.A. | Influence of carbon nanotubes on steel–concrete bond strength | *Materials and Structures* | 2018 | 20 |
| [35] | Hassan, A; Elkady, H; Shaaban, I.G. | Effect of adding carbon nanotubes on corrosion rates and steel–concrete bond | *Scientific Reports* | 2019 | 20 |
| [36] | Lee, H.; Jeong, S.; Cho, S.; Chung, W. | Enhanced bonding behavior of multi-walled carbon nanotube cement composites and reinforcing bars | *Composite Structures* | 2020 | 12 |
| [37] | Qasem, A.; Sallam, Y.S.; Hossam Eldien, H.; Ahangarn, B.H.; Eldien, H.H.; Ahangarn, B.H. | Bond-slip behavior between ultra-high-performance concrete and carbon fiber reinforced polymer bars using a pull-out test and numerical modelling | *Construction and Building Materials* | 2020 | 11 |
| [38] | Irshidat, M.R. | Improved bond behavior between FRP reinforcing bars and concrete with carbon nanotubes | *Construction and Building Materials* | 2020 | 7 |
| [39] | Song, X. B; Cai, Q.; Li, Y.Q.; Li, C.Z. | Bond behavior between steel bars and carbon nanotube modified concrete | *Construction and Building Materials* | 2020 | 4 |
| [40] | Irshidat, M.R. | Bond strength evaluation between steel rebars and carbon nanotubes modified concrete | *Case Studies in Construction Materials* | 2021 | 1 |

Hawreen and Bogas [34] performed pull-out tests of 12 mm diameter steel bars in CNT concrete, employing different dispersion techniques and CNT contents between 0.05% and 0.10% by weight of cement. The results obtained by the authors were promising, indicating that the CNT addition to concrete can improve the compressive strength and the steel–concrete bond up to 21% and 14%, respectively, compared with the reference concrete without the CNT addition.

Hassan et al. [35] studied the adherence of steel bars in CNT concrete through pull-out tests considering 12 and 16 mm bar diameters and 75 mm anchorage length for all specimens, while the CNT contents were 0.01%, 0.02%, and 0.03% by weight of cement. The results revealed that the steel–concrete bonding performed better in the CNT concrete when compared to the reference one. The addition improved the adhesion of specimens with 12 and 16 mm steel bars up to 36% and 21%, respectively.

Lee et al. [36] investigated the bonding behavior between steel bars and mortar with a CNT addition through pull-out tests. The authors tested CNT contents equal to 0.125%, 0.5%, and 1.0% by weight of cement, 10 mm diameter, and 50 mm anchorage length. The

results showed that the 0.5% CNT content increased the compressive strength by almost 16% and improved the mortar–steel bonding by approximately 13%.

Qasem et al. [37] studied the influence of a CNT addition to ultra-high-performance concrete (UHPC) through pull-out tests of steel bars and carbon fiber-reinforced polymer (CFRP) bars. They considered bar diameters equal to 12 and 16 mm and CNT contents between 0.01% and 0.10% by weight of cement. The results showed that compared to the reference concrete (without added CNT), the maximum bond strength in the samples with 12 and 16 mm diameter steel bars is about 34.7% and 48.5% higher than for similar specimens with CRFP bars.

Irshidat [38] performed pull-out tests on CNT concrete considering 14 mm diameter steel bars with different anchorage lengths, 100, 150, and 200 mm. The CNT contents tested were 0.05%, 0.10%, and 0.20% by weight of cement. Three main failure modes were observed in the tests: concrete splitting, bar pull-out, and bar breakage. The bar diameter affected the failure mode, but the CNT content or the anchorage length did not. In addition, the experimental results revealed that CNT improved the bond strength. Specifically, the 0.10% and 0.20% contents increased the bond strength between steel and concrete up to 9% and 15%, respectively.

Song et al. [39] studied the effect of different CNT contents, concrete cover (c), and bar diameter (d) in the steel–concrete bond, performing beam tests. The results indicated increases of 37.2% and 49.7% in the maximum bond strength when using CNT contents equal to 0.10% and 0.15% by weight of cement, respectively. In addition, the maximum bond strength increased linearly with the c/d ratio, and, for a fixed c/d value, varied with the CNT dosage.

Irshidat [40] investigated the effect of incorporating CNT to concrete on its mechanical properties, as well as on the steel–concrete bond behavior, considering CNT contents equal to 0.05%, 0.10%, and 0.20% by weight of cement, bar diameters equal to 12, 14, 16, and 18 mm, and anchorage lengths equal to 100, 150, and 200 mm. Through pull-out tests, the author found that the incorporation of CNT affected the steel–concrete bond behavior, improving the initial bond strength and stiffness, as well as changing the failure mode of the specimens from concrete splitting to bar pull-out. In addition, the adhesion between steel and CNT concrete improved with increasing anchorage length and decreasing bar diameter.

*3.2. Systematic Analysis*

3.2.1. Cement-Based Material

Considering the seven selected articles, six (86%) performed bond tests of steel bars in some type of concrete, with three (43%) using a conventional concrete with a compressive strength of 50 MPa [34,38,40], two (29%) also using a conventional concrete but with a compressive strength of 30 MPa [35,39], and one (14%) using an ultra-high-performance concrete with a compressive strength higher than 150 MPa [37], and all of these strengths were measured at 28 days. In this context, Hawreen et al. [34] state that the addition of CNTs to cementitious matrices is generally used in mortars or cement paste because of the difficulty of dispersing them in a mixture with coarse aggregates and mainly in large quantities. Moreover, this lower tendency to add CNTs to a concrete matrix may be related to its high cost concerning the other constituents of the mixture [41]. Still, only one work (14%) investigated the adherence of steel bars in a mortar [36].

As can be seen, the analysis of the cement-based materials studied in the selected articles shows that there is still no research on the adherence of steel bars in concrete with the addition of CNTs plus some other reinforcement, such as FRP bars, synthetic or steel fibers, as well as in concrete with the addition of CNTs and partial or total replacement of natural by artificial or recycled aggregates, such as civil construction demolition waste (CDW) or even rubber waste. It is understood, therefore, that this subject should be better investigated in future research because maybe the use of CDW contributes to the sustainability of buildings while the CNTs ensure the mechanical strength required for a structural component of reinforced concrete.

### 3.2.2. Characteristics, Dispersion Techniques, and Dosage of Carbon Nanotubes

Table 2 summarizes the main information about the CNTs employed in the selected bibliography: CNT type, aspect ratio (length/diameter), type of delivery, dispersion technique, and contents used in the mixture with the cement-based material.

**Table 2.** Characteristics, dispersion techniques, and dosage of carbon nanotubes.

| Reference | CNT Type | Aspect Ratio | Supply Type | Dispersion Technique | CNT Dosage |
|---|---|---|---|---|---|
| [34] | MWCNT (i) CNTSS (ii) CNTSL (iii) CNTPL (iv) CNTCOOH (v) CNTOH | (i) 300 (ii) 667 (iii) 667 (iv) 667 (v) 1000 | (i) Suspension (ii) Suspension (iii) Powder (iv) Powder (v) Powder | (i,ii) Magnetic stirring in water with surfactant + sonication for (i) 45 min or (ii) 30 min (iii,v) Magnetic stirring with 40% of the water, CNT and surfactant in the ratio (iii) 1:1 or (iv) 1:0.5 or (v) 1:0.25 + addition of the remaining 60% of water + magnetic stirring for 4 h + sonication for 30 min | 0% (i) 0.05% (ii) 0.05% (iii) 0.10% (iv) 0.05% (v) 0.05% |
| [35] | MWCNT | Not specified | Not specified | Dispersion in water with superplasticizer | 0%, 0.01%, 0.02%, 0.03% |
| [36] | MWCNT | Not specified | Suspension | Dispersion in distilled water and sonication at 22 Hz for 8 h | 0%, 0.125%, 0.5%, 1% |
| [37] | Not specified | Not specified | Not specified | Dispersion in water with superplasticizer | 0%, 0.01%, 0.02%, 0.03%, 0.05%, 0.10%, 0.20%, 0.50% |
| [38] | Not specified | 158 | Suspension | Dispersion with water and manual agitation + sonication (duration not specified) | 0%, 0.05%, 0.10%, 0.20% |
| [39] | MWCNT | 1000 | Suspension | Water dispersion with superplasticizer + sonication (duration not specified) | 0%, 0.05%, 0.10%, 0.15%, 0.20%, 0.50% |
| [40] | MWCNT | 158 | Suspension | Dispersion in water and manual agitation + sonication for 20 min | 0%, 0.05%, 0.10%, 0.20% |

Notes: CNTSS—unfunctionalized carbon nanotubes with shorter aspect ratio in aqueous suspension; CNTSL—unfunctionalized carbon nanotubes with longer aspect ratio in aqueous suspension; CNTPL—unfunctionalized carbon nanotubes with longer aspect ratio in the powder form; CNTCOOH—carboxyl-functionalized nanotubes with –COOH groups and higher aspect ratio; CNTOH—carboxyl-functionalized nanotubes with –OH groups and higher aspect ratio.

Regarding the CNT characteristics, in five studies (71%) MWCNTs were used, and in two (29%) this information was not specified; although, it is believed that they also chose MWCNTs. The large-scale synthesis of MWCNTs can be easily achieved by performing various advanced chemical vapor deposition (CVD) methods, which makes them easier to process, contributing to a more commercial final value compared to SWCNTs [42]. Furthermore, SWCNTs are more likely to agglomerate and are dispersed using a long dispersion route involving physical and chemical treatments, which can damage the nanotube or make the process unsuitable for industrial application [43]. In summary, the predominant use of MWCNTs in the portfolio articles occurs because they have a lower cost, which makes them more available, and less likely to agglomerate, which guarantees a more homogeneous mixture [43]. Furthermore, only one (14%) of the papers experimented with CNTs supplied in powder form in addition to suspension [34], while four (57%) employed CNTs only in aqueous suspension, and in two papers (29%) this information was not made explicit. In only one paper (14%), the CNTs employed were functionalized before dispersion [34]. Manzur et al. [44] report that greater hydrophilicity of functionalized CNTs promotes greater absorption of water available in the system, which hinders the cement hydration reactions. Consequently, smaller increases in the mechanical properties of cementitious composites with functionalized CNTs were observed compared to non-functionalized CNTs [6]. Other authors reported benefits of functionalization in dispersion [44,45], but the effectiveness of this technique is still controversial since some authors noted no difference [46]. Furthermore, only one manuscript evaluated the influence of CNTs' aspect ratios on bond strength [34], concluding that the greatest bond strength gain occurred in concrete with CNTs of a higher aspect ratio because of the better effectiveness of their bridging effect.

The most used dispersion technique was mixing the nanomaterials with water, followed by sonication. This method generally involves dispersing agents (surfactants), which prevent agglomeration and ensures the solution stability, and possibly covalent functionalization. Other methods try to disperse the CNTs on the cement particles using a non-aqueous medium [47] or growing the nanotubes directly on the cement grains [48].

As the production of cement-based materials inevitably depends on the mixing water, an aqueous medium, this dispersion technique may be the most appropriate for CNT concrete. Considering the studies that employed CNTs in an aqueous suspension, one (14%) did not perform sonication, only dispersing the CNTs in water with a superplasticizer or surfactant. Regarding the studies with powdered CNTs, only one manuscript (14%) employed a surfactant in the mixing process with water [34], precisely because the CNTs used in this case were powdery. In this aspect, it is worth noting that the combination of sonication with some dispersant can be effective because, while the first promotes a temporary dispersion of CNTs due to the permanent existence of van der Waals forces, the second prevents agglomeration and ensures the solution stability [3].

Furthermore, the CNT content in the cement matrix influences steel–concrete bonding, mechanical properties, and durability, and is directly related to good dispersion of the material. Thus, it is important to investigate up to what levels it is effective to increase this content since it is a high-cost material in the construction market. With this perspective, Figure 2 shows the frequency that contents between 0.01% and 1.00% by weight of cement were employed in the selected bibliography.

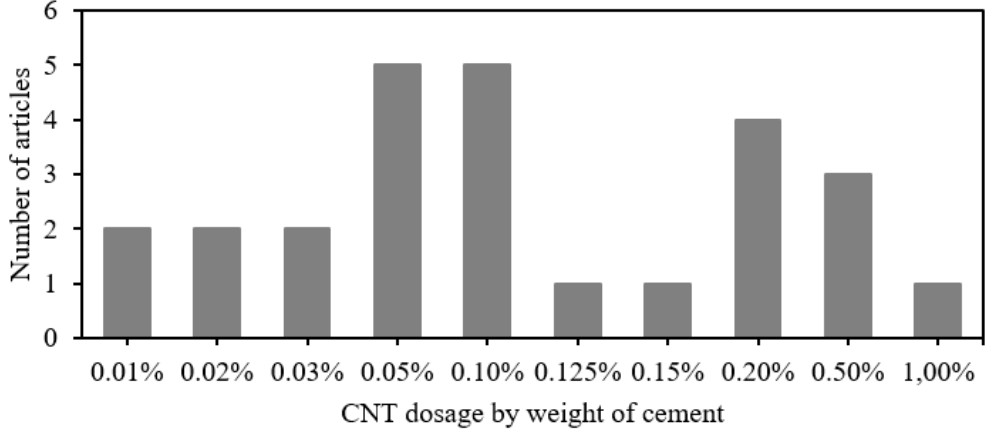

**Figure 2.** Frequency of each CNT dosage in the selected bibliography.

As shown in Figure 2, the 0.05% and 0.10% contents were the most frequent, being employed in five papers (71%), followed by 0.20%, tested in four studies (57%), and 0.50%, used in three articles (43%). Contents of 0.01%, 0.02%, and 0.03% were employed in two manuscripts (29%) each, while 0.125%, 0.15%, and 1.00% were used only once (14%) each. In this context, several researchers have studied the effects of adding CNT to different concrete types on mechanical and durability properties [49–57]. In high concentrations CNTs are more difficult to disperse and may agglomerate due to van der Walls forces, and generate large pores in the cementitious composite, which worsens its mechanical properties, and consequently, its adhesion to steel bars [58]. With this in view and further considering that CNTs are not yet produced on a large scale [43], scientific investigations should focus on contents lower than 0.10% by weight of cement.

### 3.2.3. Bond Tests and Specimen Geometry

Table 3 presents the tests performed on each article of the portfolio and their main characteristics: specimen geometry, specimen dimensions, machine and its loading capacity, and load or displacement applied to the specimen.

**Table 3.** Characteristics of bond tests and specimens.

| Reference | Bond Test | Specimen Geometry | Specimen Dimensions (mm) | Machine | Displacement or Load Rate |
|---|---|---|---|---|---|
| [34] | Pull-out test | Cubic (L × W × H) | 200 × 200 × 200 | Hydraulic Cylinder (112 kN) | 0.1 kN/s |
| [35] | Pull-out test | Cubic (L × W × H) | 150 × 150 × 150 | Universal Testing Machine (1000 kN) | Not specified |
| [36] | Pull-out test | Cubic (L × W × H) | 100 × 100 × 100 | Universal Testing Machine (1000 kN) | 1.27 mm/min |
| [37] | Pull-out test | Prismatic (L × W × H) | 200 × 200 × (10 d) | Universal Testing Machine (100 kN) | 0.8 mm/min |
| [38] | Pull-out test | Cylindrical (D × H) | 150 × 300 | Universal Testing Machine * | 0.3 mm/min |
| [39] | Beam test | Prismatic (L × W × H) | 600 × W × 240 (W not specified) | Hydraulic Actuator (500 kN) | 0.01 mm/min |
| [40] | Pull-out test | Cylindrical (D × H) | 150 × 300 | Universal Testing Machine * | 0.3 mm/min |

Notes: L—length; W—width; H—height; d—bar diameter; D—specimen diameter; * Capacity not specified.

Regarding the bond test performed, six articles (86%) performed the pull-out test and only one (14%) opted for the beam test. This is because, although the beam test represents reliably the steel–concrete bond behavior under tension [39], it is very laborious [59], while the pull-out test is simpler and easier to execute [60], provides accurate results [61], and shows clearly the influence of each variable on the test results [62]. Moreover, different setups and standards were used in the tests, considering the load or displacement applied to the specimen, the machine used, the geometry, the dimensions, and the specimen number, which reflect the lack of standards in the study of steel–concrete bonding. Song et al. [39] were the only authors who performed the beam flexural test, following the RILEM-CEB RC5 [24] and the Chinese GB/T 50,152 [63] standards. To the best of the authors' knowledge, the specimen dimensions' influence on this specific test has not yet been investigated. In pull-out tests, Hawreen and Bogas [34] used three cubic specimens per sample, with the RILEM-CEB RC6 standard [23] as reference. Hassan et al. [35] also tested three cubic specimens for each sample, but did not follow any reference standard. Lee et al. [36] used five cubic specimens per sample, following the Japanese JSCE standard [64]. Qasem et al. [37], in turn, tested four prismatic specimens per sample, without following any reference standard. Finally, Irshidat [38,40] used three cylindrical specimens per sample, but did not follow any reference standard. The author chose to use specimens with the same dimensions usually employed in compressive strength tests. At this point, it is noteworthy that there are studies in the literature evaluating the specimen characteristics' influence on the pull-out test and pointing to different experimental results when using varied geometries (cubic, prismatic, or cylindrical) [65,66].

Considering that no standard was used more than once in the articles in the portfolio and that in four manuscripts (57%) no standard was even followed, it is understood that extensive experimental studies according to some standard should be developed. Among the standards, the RILEM-CEB RC6 [23], which is commonly applied in pull-out tests of various types of cement-based materials, stands out.

### 3.2.4. Rebar Characteristics

The reinforcement most employed in reinforced concrete structural elements, carbon steel ribbed bars with a ribbed surface, were in all of the selected scientific works. Regarding the material, some authors have also investigated the bonding of polymer bars in concrete, as shown in detail in Table 4.

**Table 4.** Rebar type, diameter, and anchorage length.

| Reference | Type | Diameter | Anchorage Length |
|---|---|---|---|
| [34] | Carbon steel ribbed bar | 12 mm | 8 d |
| [35] | Carbon steel ribbed bar | (i) 12 mm<br>(ii) 16 mm | (i) 6.25 d<br>(ii) 4.70 d |
| [36] | Carbon steel ribbed bar | 10 mm | 5 d |
| [37] | Carbon steel ribbed bar<br>Carbon fiber reinforced polymer bar | 12 mm<br>16 mm | 5 d |
| [38] | (i) Carbon steel ribbed bar<br>(ii) Glass fiber reinforced polymer bar<br>(iii) Carbon fiber reinforced polymer bar | (i) 14 mm<br>(ii) 10 and 12 mm<br>(iii) 10 and 12 mm | (i) 10.7 d<br>(ii) 8.3 d, 12.5 d, 15.0 d e 16.7 d<br>(iii) 8.3 d, 12.5 d, 15.0 d e 16.7 d |
| [39] | Carbon steel ribbed bar | (i) 18 mm<br>(ii) 22 mm<br>(iii) 25 mm | (i) 4.4 d<br>(ii) 5.0d<br>(iii) 6.1d |
| [40] | Carbon steel ribbed bar | (i) 12 mm<br>(ii) 14 mm<br>(iii) 16 mm<br>(iv) 18 mm | (i) 12.5 d<br>(ii) 7.1 d, 10.7 d and 14.2 d<br>(iii) 9.4 d<br>(iv) 8.3 d |

Table 4 indicates that ribbed carbon steel bars will continue to be widely employed in reinforced concrete structures. However, new cement-based materials are constantly being developed, especially those with some kind of addition or substitution, making it necessary to better investigate the adherence of smooth steel bars in concrete. This is a fact considering that, in ribbed rebar, the mechanical interlock has a very significant influence on its bonding to the concrete, and this makes it much more difficult to analyze the effect of chemical and micro-mechanical connections. This makes it difficult to investigate, for example, what the contribution is of the addition of nanoparticles (e.g., CNTs) to the cementitious matrix of the concrete for its adherence with the reinforcing bars [67].

Diameter is one of the most evaluated parameters in the study of steel–concrete bonding because it directly influences the reinforcement surface area and, as a consequence, the portion of the frictional bond stress. Table 4 shows that the 12 mm bar diameter was the most frequent, appearing in four papers (57%), followed by 16 mm (43%), 14 and 18 mm (29%), and 10, 22, and 25 mm (14%) diameters. Therefore, it is noteworthy that thin bars (diameter of less than 10 mm) were not investigated in the selected literature. According to Carvalho et al. [26], the small number of research studies on the connection of thin bars raises doubts about the parameters used to estimate the anchorage length of these bars in reinforced concrete elements, which is proposed by the main standards. In this sense, even though there are several studies on steel–concrete bonding, there is still no specific standard for thin bars [68], regardless of the use of CNTs or even other additions. Regarding the bar diameter influence, different findings were obtained in the portfolio [35,40], which ratify that the steel–concrete bond does depend on the diameter, but evaluating it alone does not lead to an assertive conclusion. It is, then, necessary to associate it with other test parameters, such as the reinforcement cover, the concrete strength, the confinement rate, or even the type of concrete studied.

The bar anchorage length, in turn, depends on the concrete's tensile strength, the bar yield strength, the rebar's surface type, the rebars' position during concrete casting, and on the bars' diameter. When calculating the anchorage length, the Brazilian standard ABNT NBR 7480 [69], for instance, establishes a surface conformation coefficient with a minimum value of 1.0 for bars with a diameter smaller than 10 mm, regardless of the rebar's surface type, but for diameters greater than this, the standard recommends a value of 1.5 for ribbed bars. However, some studies have identified that thin bars may not meet the requirements of Brazilian standards, with conformation coefficients lower than the minimum specified [26,70]. In bonding studies, it is expected that the anchorage length ensures uniform distribution of stresses at the steel–concrete interface and is usually estimated using a diameter multiplier factor (d). In the selected bibliography,

anchorage lengths between 4.0 d and 7.5 d were adopted eight times, between 7.5 d and 11.0 d five times, and between 11.0 d and 14.5 d two times. In this sense, Ertzibengoa et al. [71] point out that the anchorage length is limited so that the bar pull-out occurs before the yielding of the steel. Nevertheless, its value depends on the test type and, consequently, on the dimensions of the specimens. Logically, larger specimens will bear longer anchorage lengths. Within this context, Carvalho et al. [26] stated that the 5 d anchorage length suggested by RILEM RC 6 [23] can be considered short for thin bars, and therefore suggested the 10 d anchorage length for these cases so that it can be comparable to that adopted in the beam test recommended by RILEM RC 5 [24]. In summary, further extensive studies on the real influence of anchorage length on the bond between steel bars and concrete with or without CNT addition should be developed. For this, the pull-out test is suggested, using the same specimen dimensions, the same concrete, and the same diameter, varying only the anchorage length.

## 4. Conclusions

A systematic review of the literature on the bonding of steel bars in concrete with the addition of carbon nanotubes, covering seven articles, was conducted in this study. From the analyses carried out, the following conclusions can be drawn:

1.  The bonding of steel bars in CNT concrete was studied in 86% of the selected articles, and in CNT mortar it was investigated in the remaining 14%. Although concrete was employed most frequently, no papers considered associating CNTs with other types of reinforcement, such as synthetic or steel fibers, or employing CNTs in concrete modified by replacement of natural by artificial or recycled aggregates, such as construction demolition waste or rubber waste. Therefore, considering the portfolio of this work, there is a gap for future research on such adherence in the mentioned composites and a need to investigate it further in CNT mortar;

2.  At least 71% of the studies employed MWCNTs. SWCNTs are often dispersed using a long dispersion route involving physical and chemical treatments, which can damage the nanotube or make its industrial application unfeasible. On the other hand, a large-scale synthesis of MWCNTs can be easily achieved by performing various advanced CVD methods, which makes them easier to process, and consequently cheaper. Thus, the choice for MWCNTs seems to be already consolidated and tends to continue in future research;

3.  Only 14% of the papers added CNTs in powder form to concrete or used functionalized CNTs before dispersion. The higher hydrophilicity of functionalized CNTs may hinder the cement hydration reactions and lead to lower gains in the mechanical properties of cementitious materials with non-functionalized CNTs. However, this technique is still controversial, as both benefits and indifference regarding its use have been reported in the literature. Therefore, the effects of functionalized CNT application on cementitious composites need to be better investigated;

4.  In the portfolio of articles in this review of the literature, mixing the nanomaterials with water followed by sonication was the most widely used dispersion technique, regardless of the type of CNT delivery. This method generally involves dispersing agents (surfactants), which prevent agglomeration and ensures the solution stability, and, possibly, covalent functionalization. Other methods try to disperse the CNTs on the cement particles using a non-aqueous medium or growing the nanotubes directly on the cement grains. As the production of cementitious composites inevitably depends on the mixing water, an aqueous medium, pre-dispersion in water followed by sonication seems to be the best option for future research involving CNT concrete;

5.  CNT contents of 0.05% and 0.10% by weight of cement were adopted in 71% of the selected manuscripts, and values above these did not lead to significant bond strength gains. Not yet industrially scaled, and therefore with high cost, in high concentrations CNTs are more difficult to disperse and may agglomerate due to van der Walls forces, and generate large pores in the cementitious composite, which worsens its mechanical

properties, and consequently, its adhesion to steel bars. With this in view, scientific investigations should focus on contents lower than 0.10%, which is in agreement with the literature;

6.  In 86% of the papers, the pull-out test was used, and in the remaining 14% the authors adopted the beam test. The comparison of the two tests indicates the difficulty of instrumentation as a disadvantage of the beam test. Like the beam test, the pull-out test also leads to accurate results, shows clearly the influence of each variable on the outcomes, and most importantly, is simple and well accepted by the scientific community. Given that, the pull-out test seems to be the best option in the investigation of steel–concrete bond behavior;

7.  Considering the wide dispersion of the setups used in pull-out tests, especially regarding the specimen geometry and dimensions, and the applied displacement or load rate, and that in 57% of the selected articles no standard was used, extensive experimental research according to at least one standard is required. The RILEM-CEB RC6 [22] standard is one of the most widely used in the study of bond behavior of steel bars in various types of cement-based materials, and is, therefore, recommended in the present work;

8.  Ribbed steel bars were employed in 95% of the articles. It is understood that they will continue to be used in studies, but the development of new types of concrete requires that the adherence of plain bars be better investigated. Furthermore, no studies in the portfolio have employed smooth steel bars in the CNT concrete specimens, which may be a better alternative to investigate the chemical effect of adding these nanomaterials for improving steel–concrete adhesion, since this disregards the influence of the mechanical interlock on the steel–concrete bond;

9.  Bars with a diameter smaller than 10 mm were not investigated in the selected bibliography. Considering that it influences the rebar surface area, and therefore, the portion of the frictional bond stress, it should be emphasized that in-depth investigations on the adherence of thin bars in concrete need to be developed, regardless of the use of CNTs or even other additions;

10. Various anchorage lengths were tried in the papers, but no clear pattern was identified in the results. As a parameter dependent on the concrete's strength, on the bar characteristics, as the diameter, on the test type, and, consequently, on the dimensions of the specimens, further extensive studies on the real influence of anchorage length on the steel–concrete bond, regardless of the use of CNTs, should be developed.

These conclusions reveal that the application of CNTs requires a full understanding of their contribution to improving steel–concrete bonding. It is emphasized that these manuscript findings are limited to the analysis of the presented portfolio of articles. Another limitation is that the influence of the type and degree of confinement, yield stress of the bar, and the concrete age were not considered in the review; factors that also need to be better understood in future research

**Author Contributions:** Conceptualization, E.D.R.; methodology, E.D.R., H.F.R. and R.C.d.A.; formal analysis, E.D.R. and H.F.R.; investigation, E.D.R.; resources, A.C.d.S.B.; data curation, E.D.R. and H.F.R.; writing—original draft preparation, E.D.R., H.F.R., P.L., R.C.d.A., F.S.J.P. and A.C.d.S.B.; writing—review and editing, P.L., R.C.d.A., F.S.J.P. and A.C.d.S.B.; visualization, P.L., R.C.d.A. and F.S.J.P.; supervision, F.S.J.P. and A.C.d.S.B.; project administration, A.C.d.S.B. All authors have read and agreed to the published version of the manuscript.

**Funding:** This research received no external funding.

**Institutional Review Board Statement:** Not applicable.

**Informed Consent Statement:** Not applicable.

**Data Availability Statement:** All of the data in this study have been listed in the paper.

**Acknowledgments:** We would like to acknowledge the Centro Federal de Educação Tecnológica de Minas Gerais (CEFET-MG), the Coordenação de Aperfeiçoamento de Pessoal de Nível Superior—Brazil (CAPES), and the Conselho Nacional de Desenvolvimento Científico e Tecnológico—Brazil (CNPq, PQ 315653/2020-5).

**Conflicts of Interest:** The authors declare no conflict of interest.

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
