# Peer review of "Bonding of Steel Bars in Concrete with the Addition of Carbon Nanotubes: A Systematic Review of the Literature"

_buildings, doi:10.3390/buildings12101626_

Round 1
Reviewer 1 Report
The paper discussed from the literature the bonding properties of steel bars in concrete with carbon nanotubes. Very limited number papers are finally considered for discussion which may lead false conclusions. The paper needs a major improvement to be appropriate for publication. Some specific comments are given as follows which should be addressed in the revised manuscript.
1. Adherence of steel bar in mortar with CNT is mentioned at page 5 lines 199- 200. However, no findings from the literature (35) are not mentioned in the manuscript.
2. Generally conclusive summary of the findings of specific properties are given in a literature review article. However, in this review manuscript these are absent for all properties of concrete with CNT.
3. It was expected from a literature review to have a section where the critical findings should be discussed. This paper is lacking this section.
4. The paper always suggested for further study for every property of the concrete. However, this type of comment should be given after discussing the current findings available in the literature.
5. The conclusions should be revised based on the revised manuscript where, some specific critical findings should be mentioned.
Several grammatical mistakes are available in the manuscript which should be correct in the revised version of the paper. Some are mentioned as follows, (a) line 18, remove the word “unpublished”, (b) lines 102-103, change words “more used” in the sentence, (c) line 256, remove “Error! Reference source not found”, (d) line 267, remove “2Error! Reference source not found”,
Reviewer 2 Report
The paper is well-written and clear but requires some revision.
There are some minor comments on the writing and major comments on the Conclusion section:
Why is this study referred to as the 'unpublished' systematic literature review? Should it be just the 'present' systematic literature review?
In the first sentence of the introduction, the phrase 'among which standout those provided by ...', should it be changed to 'among which stand out are those provided by ...'?
In lines 33 and 34 on p.1, the word higher should be changed to greater.
Figure 1 presents that a full-text analysis was done on 17 articles. It is not clear what is the scope of the review of the remaining articles. KDP
I am not sure if the phrase, 'as far as it is known', suitable in technical writing.
There were 12 research questions listed in the introduction. The conclusion covered that well but the rationale to support many conclusion points was not logical nor was it justifiable. This is a serious concern and in my opinion, requires a major revision.
For example, in Conclusion point 1, the authors stated that 86% of the selected articles studied the bonding of steel bars in CNT concrete and 14% in CNT mortar. Then, the authors stated in the next sentence "Therefore, there is a gap for future research on such adherence in concrete with CNT and other types of reinforcement...."
There are so many questions reading that. First of all, the scope of the literature review may not cover articles in those areas, secondly, the first sentence can not be used to justify the need for the research suggested in the second sentence. The word "Therefore" should not be used.
Another example is seen in Conclusion point 4, the mostly used dispersion method was concluded and then recommended that researchers should continue to use this method. There was no justification rather than because it was mostly used.
Conclusion point 5 suggested that the % CNT to be used in concrete should be lower than 0.1% because 71% of the selected manuscripts used 0.05% and 0.1%. There is not enough justification to conclude that researchers should continue to use 0.1%. The authors should state why most research in this area only used up to 0.1%, and if there is any potential to increase it. But the authors only stated that because of its high price, it is better to keep it at 0.1%, without any actual cost analysis. In many cases, the high cost of material does not mean low cost-benefit of end products. Other advantages such as improved quality, durability, and material efficiency might outweigh the material cost.
Similar comments apply to other Conclusion points, which must be revised.
There are also some more grammar and formatting errors that require checking.
Reviewer 3 Report
Manuscript Number: Buildings-1920172
A systematic review on the bonding of steel bars in concrete with the addition of carbon nanotubes is covered in the present paper. The authors also attempt to address the dispersion techniques and dosage of CNTs, bond tests and specimen geometry, and rebar characteristics. The outcome of this research might be a good source of supplementary information for further study with some major improvement before it can be recommended for publication. It is necessary to address the forwarded comments and modifications (given that they are not ranked in order of importance).
1. Page 6, line 235: The authors should add some related references that make it clear that dispersion methods don't matter.
2. Page 7, Lines 256 and 267: Formatting error.
3. Can you please critically cover the criteria for determining the length of unloaded parts for all various rebar diameters? since it also has an impact on bond-slip relationships in addition to CNTS.
4. Page 8, Line 288: Table 3: Punctuation Correction: Comma and Period on Decimal Expression.
5. Page 10, line 370: It needs some sentence and grammatical corrections.
6. The review paper will be more descriptive if it includes pictures and schematic drawings.
7. Related materials should be cited and revised in the revised manuscript.
Sample publications: There are numerous related works available.
1. https://doi.org/10.1016/j.conbuildmat.2020.119339
2. https://doi.org/10.1016/j.compstruct.2022.115642
3. https://doi.org/10.1016/j.conbuildmat.2021.124261
4. https://doi.org/10.1016/j.conbuildmat.2020.119857
5. https://doi.org/10.1016/j.compositesb.2022.109960
6. https://doi.org/10.1016/j.compstruct.2020.112201
7. https://doi.org/10.1016/B978-0-323-85856-4.00009-1
8. The conclusions (too long) need to be improved by making them more effective and please highlight the possible limitations.
9. Reference (1) correction.
Round 2
Reviewer 1 Report
The authors addressed all the comments in the revised manuscript.
Reviewer 2 Report
Thank you for submitting the revised manuscript. All the comments have been very well addressed, but the revised manuscript (in PDF) had track changes. I am not sure if the authors have submitted the clean version.
Reviewer 3 Report
This paper has been revised and may be accepted for publication.